# SAMURAI: Shape And Material from Unconstrained Real-world Arbitrary Image collections

**Mark Boss**[†]
University of Tübingen

**Andreas Engelhardt**[†]
University of Tübingen

**Abhishek Kar**
Google

**Yuanzhen Li**
Google

**Deqing Sun**
Google

**Jonathan T. Barron**
Google

**Hendrik P. A. Lensch**
University of Tübingen

**Varun Jampani**
Google

## Abstract

Inverse rendering of an object under entirely unknown capture conditions is a fundamental challenge in computer vision and graphics. Neural approaches such as NeRF have achieved photorealistic results on novel view synthesis, but they require known camera poses. Solving this problem with unknown camera poses is highly challenging as it requires joint optimization over shape, radiance, and pose. This problem is exacerbated when the input images are captured in the wild with varying backgrounds and illuminations. Standard pose estimation techniques fail in such image collections in the wild due to very few estimated correspondences across images. Furthermore, NeRF cannot relight a scene under any illumination, as it operates on radiance (the product of reflectance and illumination). We propose a joint optimization framework to estimate the shape, BRDF, and per-image camera pose and illumination. Our method works on in-the-wild online image collections of an object and produces relightable 3D assets for several use-cases such as AR/VR. To our knowledge, our method is the first to tackle this severely unconstrained task with minimal user interaction. Project page: `https://markboss.me/publication/2022-samurai/`

## 1 Introduction

Capturing high-quality 3D shapes and materials of real-world objects is essential for many graphics applications in AR, VR, games, movies, *etc*. Using active multi-view object capture setups can provide high-quality 3D assets [6, 43] but cannot scale to a large-scale set of objects that are present in the world. By contrast, image collections provided by image search results or product review images exist for nearly every object. In this work, we propose a category-agnostic technique to estimate the 3D shape and material properties of objects from such Internet image collections. Estimating 3D shapes and materials from Internet object image collections poses several challenges as the images are highly unconstrained with varying backgrounds, illuminations, and camera intrinsics. Fig. 1 (left) shows a sample image collection of an object which forms the input to our technique.

Concretely, we estimate the 3D shape and BRDF material properties [16] while also estimating per-image illumination, camera poses, and intrinsics. Several contemporary works on shape and material estimation [6, 11, 12, 52, 63] assume constant camera intrinsics, near-perfect segmentation masks as well as almost-correct camera poses given by COLMAP [49, 50]. However, it is tedious to annotate object masks in the input images manually. We also observe that COLMAP often fails due to insufficient correspondences in real-world image collections with highly varying illuminations and backgrounds even when we constrain the correspondences to lie within object masks. Instead, we

---

[†]Work done during a Student Researcher position at Google.

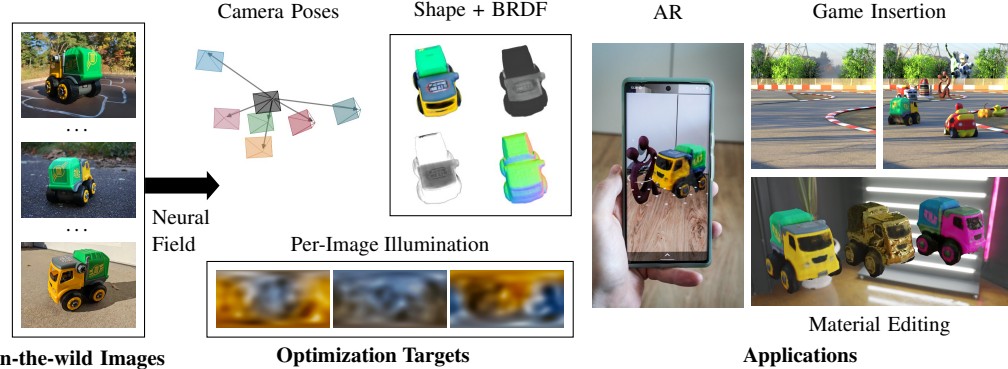

Figure 1: **Sample SAMURAI outputs and applications.** Sample input collection and the outputs on a challenging real-world unconstrained image collection. We extract meshes with material properties from the learned volumes enabling several applications in AR/VR, material editing etc.

use a rough quadrant-based pose initialization, *e.g.* (Front, Above, Right), (Front, Below, Left) *etc.*, as in NeRS [61], which usually takes only a few minutes of annotation time per image collection. We base our technique on the recent Neural-PIL [12] method that proposes to learn illumination priors along with a novel pre-integration illumination network for estimating a neural volume with 3D shape, BRDF, and per-image illumination. Neural-PIL [12] assumes perfect camera poses and the same camera intrinsics across images. Given that traditional camera pose estimations like COLMAP may fail on in-the-wild images, we propose SAMURAI to jointly optimize camera poses as well as intrinsics with a carefully designed optimization protocol (Fig. 1). Furthermore, Neural-PIL requires perfect object masks, whereas we leverage automatically estimated object masks and deal with noisy masks using a posterior scaling loss. Some key distinguishing features of SAMURAI include:

- *Flexible camera parametrization for varying distances.* Standard techniques such as NeRF [40] assume fixed near/far clipping planes with equidistant cameras to the object. In contrast, we define the neural volume in global coordinates and propose to learn clipping planes per image.

- *Camera multiplex optimization.* Optimizing a single camera per image is prone to getting stuck in local minima. We propose using a multiplex camera where we optimize several camera poses per image and then phase out the incorrect poses throughout the optimization. Although camera multiplexes are previously used in mesh optimization [21], optimizing camera multiplex with neural volumes is challenging due to inefficient ray-based neural volume rendering.

- *Posterior scaling of input images.* As different input images have different noise characteristics (*e.g.*, noisy masks), some images would be more useful for the optimization. We propose to use posterior scaling of input images that weighs the influence of different images on the optimization.

- *Mesh extraction.* We extract explicit meshes with BRDF texture from the learned neural volume making the resulting 3D models readily usable in existing graphics engines.

We observe that existing related datasets such as NeRD-dataset [11] do not capture the variations present in image collections in the wild. For instance, NeRD-dataset images have non-varying background making it easier for COLMAP to work. In addition, the illumination variations are more drastic in internet images captured by different people/cameras and at different times. To evaluate the practical in-the-wild setting, we collected image collections with 8 objects in which each image is captured under unique background and illumination conditions. In addition, we also vary the cameras used for capturing the images. Experiments on our new image collections and existing datasets demonstrate better view synthesis and relighting results with SAMURAI compared to existing works. In addition, explicit mesh extraction allows for seamless use of learned 3D assets in graphics applications such as object insertion in AR or games and material editing *etc*. Fig. 1 (right) shows some sample application results with 3D assets estimated using SAMURAI.

## 2 Related works

**Neural fields** encode spatial information in the MLP network weights, and we can retrieve the information by simply querying the coordinates [15, 39, 45, 53]. With these MLPs, we can store

alpha or density values and then explicitly render the volume using ray marching [36]. Recent works such as NeRF [40] leverage this neural volume rendering to achieve photo-realistic view synthesis results with view-dependent appearance variations. Rapid research in neural fields followed, which alternated the surface representations [44, 55], provided general improvements to the method [4, 54], reduced the long training times [14, 35, 41, 48, 56] and inference times [11, 22, 26, 35, 41, 60], enabled extraction of 3D geometry and materials [11, 42, 61], added generalization capabilities [13, 56, 65] or enabled relighting of scenes [5, 11, 12, 27, 37, 52, 63, 64]. However, most methods rely on COLMAP poses, which can fail in complex settings, such as varying illumination and locations.

**Joint camera and shape estimation** is a highly ambiguous task. An accurate shape reconstruction is only possible with accurate poses and vice versa. Often techniques rely on correspondences across images to estimate camera poses [49, 50]. Recently, several methods combined camera calibration with a joint neural volume training. Jeong *et al*. [24] (SCNeRF) rely on correspondences, and BARF [34] proposes a coarse to fine optimization using a varying number of Fourier frequencies and requires rough camera poses and NeRF-- [57] requires training the neural volume twice while keeping the previous camera parameter optimization. GNeRF [38] proposes to use a discriminator on randomly sampled views to learn a pose estimation network on synthesized views jointly. Over time the pose estimation network can estimate the real camera poses, which can then be used for the full neural volume training. A concurrent work LASSI [58] optimizes articulated shapes and camera poses for animal image collections via part discovery, but LASSIE do not estimate illumination or material properties. NeRS [61] deforms a sphere to a specific shape using coordinate-based MLPs and converts the deformation field to a mesh; while also optimizing camera poses and single illumination. Compared to previous work, our method does not rely on correspondences (*vs*. SCNeRF), which might be hard to obtain in varying illuminations, can have extremely coarse poses due to the camera multiplexing (*vs*. BARF), works in multiple Illumination (*vs*. all prior art), does not require training twice (*vs*. NeRF--) or a GAN-style training (*vs*. GNeRF).

**BRDF and illumination estimation** is a challenging ambiguous research problem. One needs controlled laboratory capture setups for high-quality BRDF capture [3, 9, 28–30]. Casual estimation enables on-site material acquisition with simple cameras and a co-located camera flash. These techniques often constrain the problem to planar surfaces with either a single shot [1, 8, 17, 23, 32, 47], few-shot [1] or multi-shot [2, 9, 18–20] captures. This casual capture setup can also be extended to a joint BRDF and shape reconstruction [5–7, 10, 25, 43, 47, 62] or entire scenes [33, 51]. Most of these methods require a known active illumination like a co-located flash. Recovering a BRDF under unknown passive illumination is significantly more challenging and ambiguous as it requires disentangling the BRDF from the illumination. Often the specular parameter is constrained to be non-spatially varying or omitted [31, 59, 63]. Recently, neural field-based decomposition achieved decomposition of scenes under varying illumination [11, 12] or fixed illumination [63], but require known, near-perfect camera poses. This can fail on challenging datasets, and our method enables the decomposition of these in-the-wild datasets.

## 3  Method

**Problem setup.** The input is a collection of $q$ object images $C_j \in \mathbb{R}^{s_j \times 3}; j \in \{1, \ldots, q\}$ captured with different backgrounds, cameras and illuminations; and can also have varying resolutions. We denote the value of a specific pixel as $C^s$. In addition, we roughly annotate camera pose quadrants with 3 simple binary questions: Left *vs*. Right, Above *vs*. Below, and Front *vs*. Back. We automatically estimate foreground segmentation masks $M_j \in \{0, 1\}^{s \times 1}$ using $U^2$-Net [46], which can be imperfect. Given these, we jointly optimize a 3D neural volume with shape and BRDF material information along with per-image illumination, camera poses, and intrinsics. This practical capture setup allows the conversion of most 2D image collections into a 3D representation with little manual work. The rough pose quadrant annotation takes about a few (3-5) minutes for a typical 80 image collection. At each point $x \in \mathbb{R}^3$ in the 3D neural volume $\mathcal{V}$, we estimate the BRDF parameters for the Cook-Torrance model [16] $b \in \mathbb{R}^5$ (basecolor $b_c \in \mathbb{R}^3$, metallic $b_m \in \mathbb{R}^1$, roughness $b_r \in \mathbb{R}$), unit-length surface normal $n \in \mathbb{R}^3$ and volume density $\sigma \in \mathbb{R}$. We also estimate the latent per-image illumination vectors $z_j^l \in \mathbb{R}^{128}; j \in \{1, \ldots, q\}$ used in Neural-PIL [12]. We also estimate per-image camera poses and intrinsics, which we represent using a 'look-at' parameterization that we explain later. Next, we provide a brief overview of prerequisites: NeRF [40] and Neural-PIL [12].

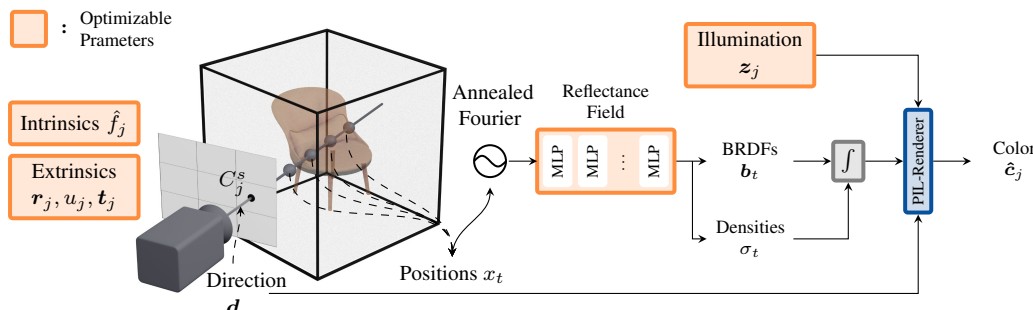

Figure 2: **Overview.** We jointly optimize the intrinsic ($\hat{f}_j$) and extrinsic camera ($\boldsymbol{r}_j, u_j, \boldsymbol{t}_j$) parameters alongside the shape ($\sigma$) and BRDF ($\boldsymbol{b}$) in a *Reflectance Field* and per-image illumination ($\boldsymbol{z}_j$). The shape is encoded in the density $\sigma$ and also used to integrate all BRDFs along a ray in direction $\boldsymbol{d}$. The composed BRDF is then rendered using Neural-PIL [12] in a deferred rendering-style.

**Brief overview of NeRF [40].** NeRF [40] models a neural volume for novel view synthesis with two Multi-Layer-Perceptrons (MLP). The MLPs take 3D location $\boldsymbol{x} \in \mathbb{R}^3$ and view direction $\boldsymbol{d} \in \mathbb{R}^3$ as input and outputs a view-dependent output color $\boldsymbol{c} \in \mathbb{R}^3$ and volume density $\sigma \in \mathbb{R}$. These output colors for a target view are computed by casting a camera ray $r(t) = \boldsymbol{o} + t\boldsymbol{d}$ into the volume, with ray origin $\boldsymbol{o} \in \mathbb{R}^3$ and view direction $\boldsymbol{d}$. The final color is then approximated via numerical quadrature of the integral: $\hat{\boldsymbol{c}}(\boldsymbol{r}) = \int_{t_n}^{t_f} T(t)\sigma(t)\boldsymbol{c}(t)\,dt$ with $T(t) = \exp(-\int_{t_n}^{t} \sigma(t)\,dt)$, using the near and far bounds of the ray $t_n$ and $t_f$ respectively [40]. The first MLP is trained to learn a coarse representation by sampling the volume in a fixed sampling pattern along each ray. A finer sampling pattern is created using the coarse density distribution in the coarse MLP, placing more samples on high-density areas. The second MLP is then trained using both fine and coarse sampling.

**Brief overview of Neural-PIL [12].** Similar to NeRF [40], Neural-PIL [12] also uses two MLPs to learn a neural volume. The first MLP is similar to NeRF with an additional GLO (generative latent optimization) embedding that models the changes in appearances (due to different illuminations) across images. In the second MLP, breaking with NeRF, Neural-PIL predicts not only a view-dependent output color but also BRDF parameters at each 3D location. The second MLP takes 3D location as input and outputs volume density and BRDF parameters (diffuse, specular, roughness, and normals). Unlike NeRF, this second MLP does not take the view direction as input but can model view-dependent and relighting effects in the rendering due to explicit BRDF decomposition. A key distinguishing factor of Neural-PIL is the use of latent illumination embeddings and a specialized illumination pre-integration (PIL) network for fast rendering, which we refer to as 'PIL rendering'. More concretely, Neural-PIL optimizes per-image illumination embedding $\boldsymbol{z}_j^l$ to model image-specific illumination. The rendered output color $\hat{\boldsymbol{c}}$ is equivalent to NeRF's output $\boldsymbol{c}$, but due to the explicit BRDF decomposition and illumination modeling, it enables relighting.

### 3.1 SAMURAI - Joint optimization of shape, BRDF, cameras, and illuminations

The main limitations of Neural-PIL include the assumption of near-perfect camera poses and the availability of perfect object segmentation masks. We observe that COLMAP either produces incorrect poses or completely fails due to an insufficient number of correspondences across images when the backgrounds and illuminations are highly varying across the image collection. In addition, camera intrinsics could vary across image collections, and the automatically estimated object masks could also be noisy. We propose a technique (we refer to as 'SAMURAI') for joint optimization of 3D shape, BRDF, per-image camera parameters, and illuminations for a given in-the-wild image collection. This is a highly under-constrained and challenging optimization problem when only image collections and rough camera pose quadrants are given as input. We address this highly challenging problem with carefully designed camera parameterization and optimization schemes.

**Architecture overview.** A high-level overview of SAMURAI architecture is shown in Fig. 2, which mostly follows the architecture of Neural-PIL [12] and NeRD [11]. However, we do not use a coarse network for efficiency reasons and only use the fine or decomposition network with an MLP with 8 layers of 128 features. Overall we sample 128 points along the ray in fixed steps. The latent embedding from the main network is interpreted with a linear layer outputting 1 feature for the density and an MLP with a hidden layer for the view and appearance-conditioned radiance. We also leverage a

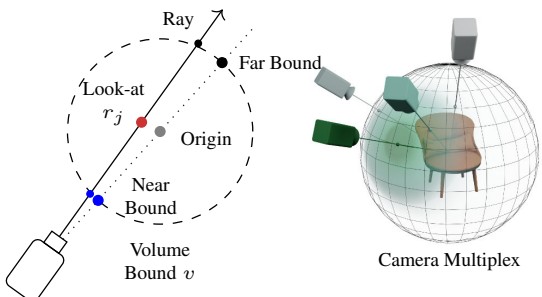
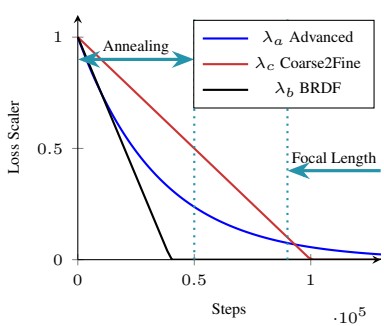

Figure 3: **Ray parametrization and Camera Multiplex.** (Left) Our ray bounds are defined by a world space sphere. The distance from the origin to the near and far points of the sphere define the sampling range. By defining a globally consistent sampling range our cameras can be placed in arbitrary distances. (Right) When optimizing multiple camera hypotheses, only the best camera should optimize the shape and appearance, here visualized in a deep green. Cameras which are not aligned that well are visualized in off-white and cannot influence the reflectance field. When the non-aligned camera poses improve during the training, they may become relevant for the network optimization.

Figure 4: **Optimization scheme.** During optimization, we use a gradual change in the loss weights using three $\lambda$ variables. Additionally, we perform a Fourier frequency annealing in the first phase of the training and delay the training of the focal length for later stages in the training. The BRDF estimation is mainly regulated by the $\lambda_b$ parameter.

BRDF decoder similar to NeRD [11], which first compresses the feature output of the main network to 16 features and compresses the embedding further to the BRDF (base color, metallic, roughness). We encourage sparsity in the embedding space using: $\mathcal{L}_{\text{Dec Sparsity}} = 1/N \sum_i^N |e_i|$, an $\mathcal{L}_1$-Loss on the BRDF embedding $e$ and a smoothness loss $\mathcal{L}_{\text{Dec Smooth}} = 1/N \sum_i^N |f(\theta; e_i) - f(\theta; e_i + \varepsilon)|$, where $N$ denotes the number of random rays, $f(\theta)$ the BRDF decoder with the weights $\theta$ and $\varepsilon$ is normal distributed Gaussian noise with a standard deviation of $0.01$. Similar to NeRD, we also predict a regular direction-dependent radiance $\tilde{c}$ in the early stages of the training. This is mainly used for training stabilization in the early stages. As this direct color prediction is only used in the early stages, we omitted it for simplicity in Fig. 2. Inspired by Tancik *et al.* [53], we add a Gaussian distributed noise to the Fourier embedding. However, as we also leverage BARF's [34] Fourier annealing, we add these random frequencies as offsets to the logarithmically spaced frequencies. Without these random offsets, artifacts from the axis-aligned frequencies, like stripes, can occur. Further details are available in the supplementary material.

**Rough camera pose quadrant initialization.** We observe that camera pose optimization is a highly non-convex problem and tends to get stuck quickly in local minima. To combat this, we propose to annotate camera pose quadrants with 3 simple binary questions: Left *vs.* Right, Above *vs.* Below, and Front *vs.* Back. This only takes about 4-5 minutes for our typical 80 image collection. Note that our pose quadrant initialization is much noisier than adding some noise around GT camera poses as in some related works such as NeRF-- [57]. This rough pose initialization is in line with recent works such as NeRS [61] that also use rough manual pose initialization.

**Flexible object-centric camera parameterization for varying camera distances.** We define the trainable per-image camera parameters using a 'look-at' parameterization with a 3D look-at vector $r_j \in \mathbb{R}^3; j \in \{1, \ldots, q\}$, a scalar up rotation $u_j \in \mathbb{R}[-\pi, \pi]$ and a 3D camera position $t_j \in \mathbb{R}^3$ as well as a focal length $f_j \in \mathbb{R}$. Furthermore, these are stored as offsets to the initial parameters, enabling easier regularization. We additionally store the offset vertical focal lengths $f_j$ in a compressed manner similar to NeRF--: $\hat{f}_j = \sqrt{f_j/h}$ [57], where $h$ is the image height in pixels. The cameras are initialized based on the given pose quadrants and an initial field of view of 53.13 degrees. We optimize a perspective pinhole camera with a fixed principal point but per-image focal lengths. The cameras are not always equidistant to the object for the in-the-wild image collections. To account for variable camera-object distances, we do not set fixed near and far bounds for each ray which is a standard practice in neural volumetric optimizations such as NeRF [40]. Instead, we define a sampling range based on the camera distance to origin, *e.g.* the near bound is $|o| - v$ and the

far bound is $|\boldsymbol{o}| + v$, where $v$ is defined as our sampling radius with a diameter of 1. We illustrate this sphere with near and far bounds in Fig. 3. This explicit computation of near and far bounds for each ray enables placing the cameras at arbitrary distances from the object. This is not possible with the existing neural volume optimization techniques that use fixed near and far bounds for each camera ray. The cameras are then placed based on the quadrants and placed at a distance to make the entire neural volume $v$ visible. This look-at parameterization is more flexible for optimizing object-centric neural volumes than more commonly used 3D rotation matrices.

**Camera multiplexes.** We observe that camera pose optimization gets stuck in local minima even with rough quadrant pose initialization. To combat this, inspired by mesh optimization works (*e.g.* [21]), we propose to optimize a camera multiplex with 4 randomly jittered poses around the quadrant center direction for each image. Optimizing multiple cameras per image would reduce the number of rays we can cast in a single optimization step due to memory and computational limitations. This makes camera multiplex optimization noisy and challenging in learning neural volumes. We propose techniques to make camera multiplex learning more robust by dynamically re-weighing the loss functions associated with different cameras in a multiplex during the optimization. This process is visualized in Fig. 3. Specifically, we compute the mask reconstruction loss $\mathcal{L}_{\text{Mask}}{}_j^i \in \mathbb{R}$ associated with each camera $i$ and image $j$. We then re-weigh each camera loss in a multiplex with $S_j = softmax(-\lambda_s \mathcal{L}_{\text{Mask}}{}_j^i)$, where $S_j \in \mathbb{R}^4$ and $\lambda_s$ is a scalar that is gradually increased during the optimization. That is, we re-weigh the loss with $\mathcal{L}_{\text{Network}\,j} = \sum_i S_j^i \mathcal{L}_{\text{Network}}{}_j^i$. This dynamic re-weighing reduces the influence of bad camera poses while learning the shape and materials. Since we can only render a random set of rays within each batch, we update the camera multiplex weights $S_j$ with a memory bank and momentum across the batches. See the supplements for more details.

**Posterior scaling of input images.** Some images are noisier than others (*e.g.*, due to camera shake) or noisy object masks $M_j$. To be robust against such noisy data, we propose to re-weigh images in the given collection. We keep a circular buffer of around 1000 elements with the recent mask losses and rendered image losses with multiplex scaling applied. We use this buffer to calculate the mean $\mu_l$ and standard deviation $\sigma_l$ of these losses. Given the recent loss statistics we also create a loss scalar using: $R_j = \max(\tanh\left(\frac{\mu_l - (\mathcal{L}_{\text{Mask}\,j} + \mathcal{L}_{\text{Image}\,j})}{\sigma_l}\right) + 1, 1)$. In a similar way to the camera posterior scaling, we employ it on a per-image basis using: $\mathcal{L}_{\text{Network}\,j} = R_j\ \mathcal{L}_{\text{Network}\,j}$.

### 3.2  Losses and Optimization

**Image reconstruction loss** is a Chabonnier loss: $\mathcal{L}_{\text{Image}}(g, p) = \sqrt{(g - p)^2 + 0.001^2}$ between the input color from $C$ for pixel $s$ and the corresponding predicted color of the networks $\tilde{c}$. We additionally calculate the loss with the rendered color $\hat{c}$.

**Mask losses** consist of two terms. One is the binary cross-entropy loss $\mathcal{L}_{\text{BCE}}$ between the volume-rendered mask and the estimated foreground object mask. The second one is the background loss $\mathcal{L}_{\text{Background}}$ from NeRD [11], which forces all samples for rays cast towards the background to be 0. We combine these losses as the mask loss: $\mathcal{L}_{\text{Mask}} = \mathcal{L}_{\text{BCE}} + \mathcal{L}_{\text{Background}}$

**Regularization losses** We compute the gradient of the density to estimate the surface normals. We use the normal direction loss $\mathcal{L}_{\text{ndir}}$ from [54] to constrain the normals to face the camera until the ray reaches the surface. This helps in obtaining sharper surfaces without cloud-like artifacts.

**BRDF losses.** The task of joint estimation of BRDF and illumination is quite challenging where the illumination can fall into a local minimum. For example, the object can be tinted in a bluish color, and the illumination is an orange color to express a more neutral color tone. As our image collections have multiple illuminations, we can force the base color $\boldsymbol{b}_c$ to replicate the pixel color from the images. This way, a mean color over the dataset is learned and prevents falling into the local minima. We leverage the Mean Squared Error (MSE) for this: $\mathcal{L}_{\text{Init}} = \mathcal{L}_{\text{MSE}}(\boldsymbol{C^s}, \boldsymbol{b}_c)$. Additionally, we find that a smoothness loss $\mathcal{L}_{\text{Smooth}}$ for the normal, roughness and metallic parameters similar to the one used in UNISURF [44] further regularizes the solution.

**Overall network and camera losses.** The final loss to optimize the decomposition network is then defined as $\mathcal{L}_{\text{Network}} = \lambda_b \mathcal{L}_{\text{Image}}(\boldsymbol{C^s}, \tilde{\boldsymbol{c}}) + (1 - \lambda_b)\mathcal{L}_{\text{Image}}(\boldsymbol{C^s}, \hat{\boldsymbol{c}}) + \mathcal{L}_{\text{Mask}} + \lambda_a \mathcal{L}_{\text{Init}} + \lambda_{\text{ndir}}\mathcal{L}_{\text{ndir}} + \lambda_{\text{Smooth}}\mathcal{L}_{\text{Smooth}} + \lambda_{\text{Dec Smooth}}\mathcal{L}_{\text{Dec Smooth}} + \lambda_{\text{Dec Sparsity}}\mathcal{L}_{\text{Dec Sparsity}}$. Here, $\lambda_b$ and $\lambda_a$ are the optimization scheduling variables. Furthermore, the camera posterior scaling is applied to these losses. For the camera optimization, we leverage the same losses as described in $\mathcal{L}_{\text{Network}}$. However, as the camera

should constantly be optimized, we do not apply posterior scaling to the losses when optimizing cameras. This allows cameras to leave the local minima. Here, we only calculate the mean loss, and therefore badly initialized camera poses can still recover over the training duration. Additionally, we define an $\mathcal{L}_1$ loss on the look-at vector $\boldsymbol{r}_j$ to constrain the camera pose to look at the object. The loss is defined as $\mathcal{L}_{\text{lookat}}$. We also use a volume padding loss, which prevents cameras from going too far into our volume bound $v$: $\mathcal{L}_{\text{Bounds}} = \max((v - |\boldsymbol{t}_j|)^2, 0)$.

**Optimization scheduling.** Fig. 4 shows the optimization schedule of different loss weights. We use three fading $\lambda$ variables to transition the optimization schedule smoothly. The $\lambda_c$ is mainly used to increase image resolution and reduce the number of active multiplex cameras. The direct color $\tilde{c}$ optimization is faded to the BRDF optimization using $\lambda_b$ and some losses are scaled by $\lambda_a$ as defined earlier. Furthermore, we perform the BARF [34] frequency annealing in the early stages of the training and delay the focal length optimization to the later stages of the training. We use two different optimizers. The networks are optimized by an Adam optimizer with a learning rate of 1e-4 and exponentially decayed by order of magnitude every 300k steps. The camera optimization is performed with a learning rate of 3e-3 and exponentially decayed by order of magnitude every 70k steps. Further details of the optimization schedule are available in the supplementary material.

| Method | Poses Not Known (10) | | Poses Available (5) (Used in NeRD/Neural-PIL) | | | |
|---|---|---|---|---|---|---|
| | PSNR↑ | SSIM↑ | PSNR↑ | SSIM↑ | Translation↓ | Rotation °↓ |
| BARF-A | 16.9 | 0.79 | 19.7 | 0.73 | 23.38 | 2.99 |
| **SAMURAI** | **23.46** | **0.90** | **22.84** | **0.89** | **8.61** | **0.86** |
| NeRD [11] | — | — | 26.88 | 0.95 | — | — |
| Neural-PIL [12] | — | — | 27.73 | 0.96 | — | — |

Table 1: **Novel View Synthesis on varying illumination datasets.** We split our datasets into those where we have poses, and highly challenging ones where the poses were not recoverable with classical methods. SAMURAI achieves considerably better performance compared to BARF-A. For reference, we also show the metrics from NeRD and Neural-PIL which require GT poses and do not work on images with unknown poses.

| Method | Pose Init | PSNR↑ | SSIM↑ | Translation↓ | Rotation °↓ |
|---|---|---|---|---|---|
| BARF [34] | Quadrant | 14.96 | 0.47 | 34.64 | 0.86 |
| GNeRF [38] | Random | 20.3 | 0.61 | 81.22 | 2.39 |
| NeRS [61] | Quadrant | 12.84 | 0.68 | 32.27 | 0.77 |
| **SAMURAI** | Quadrant | 21.08 | **0.76** | 33.95 | **0.71** |
| NeRD [11] | GT | 23.86 | 0.88 | — | — |
| Neural-PIL [12] | GT | 23.95 | 0.90 | — | — |

Table 2: **Novel View Synthesis on single illumination datasets.** For two scenes under single illumination the poses are easily recoverable. Furthermore, we can now compare with GNeRF which does not require pose initialization. As seen our method achieves a good performance. For reference the view synthesis metrics with NeRD and Neural-PIL that use GT known poses are also shown.

Figure 5: **SAMURAI-Dataset Novel Views.** Here, we show novel views and illuminations from the SAMURAI datasets alongside our reconstruction. The illumination conditions are accurately reproduced.

| Method | PSNR↑ | SSIM↑ |
|---|---|---|
| w/o Camera Multiplex | 23.01 | 0.87 |
| w/o Posterior Scaling | 23.51 | 0.88 |
| w/o Annealing/Resolution | 22.73 | 0.83 |
| w/o Random Fourier Offsets | 24.01 | 0.91 |
| w/o Regularization | 21.77 | 0.86 |
| **Full** | **24.31** | **0.92** |

Table 3: **Ablation study.** view synthesis and relighting results on two scenes (Garbage Truck and NeRD car) show that ablating any of the proposed aspects of SAMURAI can result in worse results demonstrating their importance.

## 4 Experiments

**Datasets** For evaluations, we created new image collections of 8 objects (each with 80 images) captured under unique illuminations and locations and a few different cameras. We refer to this dataset as the SAMURAI dataset. The images reflect the practical and challenging input scenario we are targeting in this work. Common methods such as COLMAP fail to estimate correspondences and camera poses for this dataset. Therefore, we cannot run methods that require poses on this dataset. Additionally, we evaluate on 2 CC-licensed image collections from online sources of the statue of liberty and a chair. We also use the 3 synthetic and 2 real-world datasets of NeRD [11] under varying illumination, where poses are available. Lastly, to showcase the performance with other methods, we use the 2 real-world datasets from NeRD, which are taken under fixed illumination. In total, we

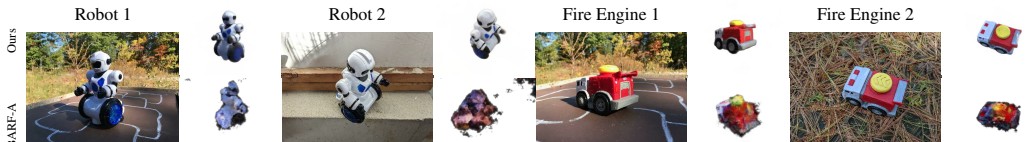

Figure 6: **Comparison with BARF-A.** When comparing novel view synthesis and relighting results of SAMURAI (top) with BARF-A (bottom), SAMURAI produces more accurate camera poses and captures the object better. BARF-A is sometimes unable to recover the shape and poses.

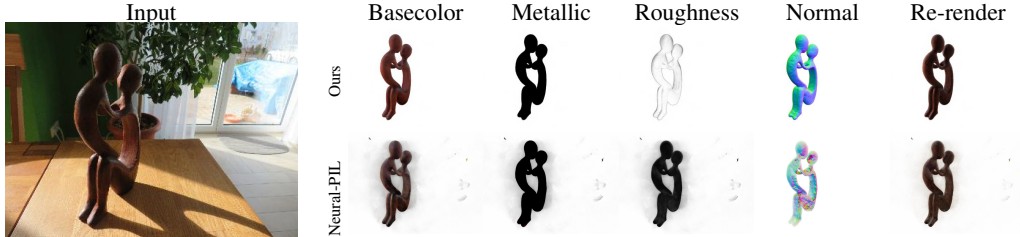

Figure 7: **Comparison with Neural-PIL decomposition.** Notice our accurate pose alignment, plausible geometry, reduced floaters and accurate BRDF decomposition, compared to Neural-PIL, even without relying on near perfect poses.

evaluate SAMURAI on 17 scenes. Please refer to the supplementary material for an overview of the SAMURAI datasets along with other datasets we experimented with.

**Baselines.** Currently, there exists no prior art that can tackle varying illumination input images while jointly estimating camera poses. So, we compare with a modified BARF [34] technique, which can store per-image appearances in a latent vector. We call this baseline BARF-A. Additionally, on scenes with fixed illumination, we can compare with GNeRF [38], the regular BARF, and a modified version of NeRS [61] (details in the supplement). On the datasets where poses are easily recovered or given, we can also compare with NeRD [11] and Neural-PIL [12], which require known, near-perfect camera poses. We provide BARF, BARF-A, and NeRS with the same pose initialization used in SAMURAI.

**Evaluation.** We perform novel view synthesis using learned volumes and use standard PSNR and SSIM metrics w.r.t. ground-truth views. We held out every 16th image for testing. We optimize the cameras and illuminations on the test images for evaluation purposes but do not allow the test images to affect the main decomposition network or camera training. Additionally, we perform the Procrustes analysis on the recovered camera poses to evaluate the pose optimization.

**Results on varying illumination datasets.** We divide the varying illumination datasets into the ones without GT poses, and those with accurate camera poses (either via GT or COLMAP). Since these datasets have varying illumination, we need to perform both view synthesis and relighting to obtain target views. Table 1 shows the metrics computed w.r.t. test views. Results show that we considerably outperform BARF-A in both PSNR and SSIM metrics while at the same time solving a more challenging BRDF decomposition task. Visual results in Fig. 6 also clearly demonstrate better view synthesis and relighting results compared to BARF-A. BARF-A fails to align the camera poses, whereas SAMURAI achieves more accurate camera poses and drastically improved reconstruction quality. Only slightly perturbed poses are leveraged as starting positions in the original BARF method. Our coarse pose initialization is too noisy for the method to work accurately. SAMURAI overcomes this issue with the camera multiplex and other optimization strategies.

Fig. 7 shows the visual comparison of BRDF decompositions on MotherChild dataset from NeRD [11] along with the corresponding results from Neural-PIL [12]. In general, our method can decompose the scene even with unknown camera poses. SAMURAI also produces fewer floating artifacts and creates a more coherent surface. The roughness parameter is also more plausible in our result, as the object is rough, whereas Neural-PIL estimated a near mirror-like surface. Further results from the SAMURAI dataset are shown in Fig. 5. We show novel views and relighting results w.r.t the target test views. Visual results clearly show that SAMURAI can recover the pose and provide a consistent illumination w.r.t the ground-truth target views.

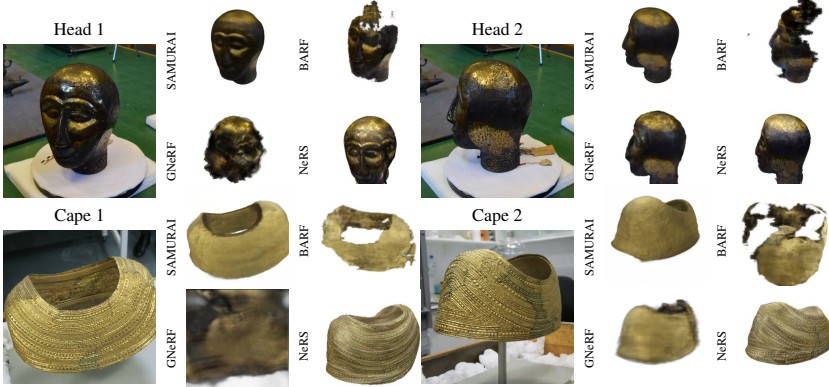

Figure 8: **Comparison with Baselines.** When comparing SAMURAI with the baselines (GNeRF, BARF, and NeRS) ours outperforms all methods in reconstruction quality and pose estimation.

**Results on fixed illumination datasets.** For image collections captured under fixed illumination, we can compare with more techniques. We compare with GNeRF, the default BARF and NeRS. We additionally can compare with Neural-PIL [12] and NeRD [11] on the near-perfect camera poses recovered from COLMAP. Results in Table 2 show that SAMURAI outperforms the baselines BARF, GNeRF, and NeRS and is also close to Neural-PIL and NeRD that uses GT camera poses. GNeRF does not require a rough pose initialization. Overall our method also achieves a good pose recovery, where NeRS only slightly outperforms our method in the translational error due to some outliers in our case. These outliers do not degrade our reconstructions due to our image posterior loss.

Fig. 8 shows sample view synthesis results of SAMURAI, BARF, GNeRF, and NeRS on sample single illumination datasets. Visuals indicate better results with SAMURAI compared to GNeRF and BARF. NeRS seems to capture more apparent detail, but the general decomposition quality is significantly better in our method, where the cape gold material is represented more accurately. NeRS also introduces a misaligned face texture in the Head scene, where two faces are visible.

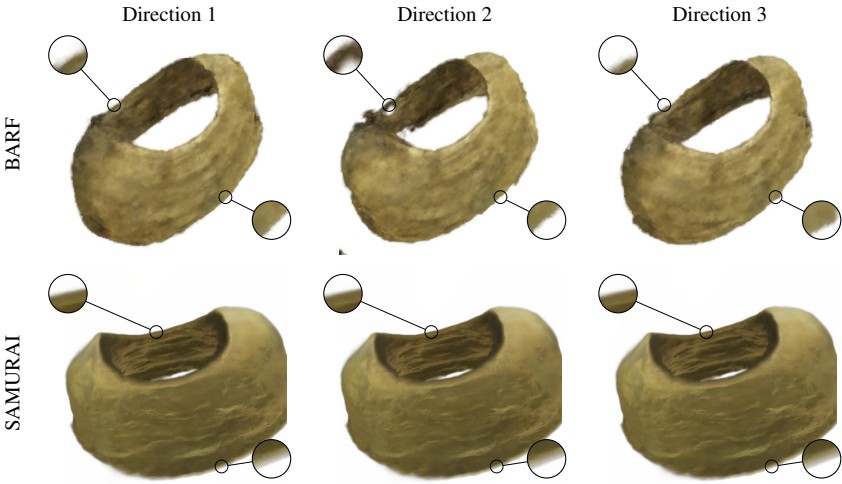

Figure 9: **BARF directional conditioning entanglement.** BARF's radiance output is conditioned based on the view direction. When we manipulate the view direction while keeping the camera static, we can see that the embedding is still entangled with the pose. BARF improves the photometric training error of unaligned poses by shifting the texture. SAMURAI instead estimates view-independent BRDFs, where this entanglement is not possible. Here, we provide magnifications on edges where this entanglement is noticeable. Due to different camera coordinate systems and optimizations, we provide similar views from SAMURAI.

**Entanglement of View direction and Radiance Conditioning in BARF.** Based on the view direction, BARF conditions the output radiance similar to NeRF [40]. In Fig. 9, we show the effect of a fixed camera with varying directional conditioning of the radiance. The texture starts to shift on the surface. BARF reduced the photometric training error of unaligned poses by slightly shifting the texture for these views. This can result in shifting textures in novel view synthesis. With the shifting textures, the shape representation worsens, and the pose alignment quality is further limited.

Our BRDF decomposition creates a view-independent texture representation, and our illumination is global and does not influence the static texture information. With this representation shifting the textures is impossible, and the texture can only remain static. Especially in pose alignment, this is highly beneficial as each camera pose has to align to a globally static model. This also explains our improvement in quality in novel view synthesis in Table 1 and Table 2. In the supplementary, we also show the effects of appearance conditioning in BARF-A.

**Ablation Study.** We perform an ablation study where we ablate different aspects of the SAMURAI model to analyze their importance. Table 3 shows the novel view synthesis average metrics on the Garbage Truck collection from the SAMURAI dataset and the synthetic car dataset from NeRD [11]. Metrics show that regularization and the coarse-to-fine optimization scheme (Fourier annealing and resolution) are the most significant contributing factors to the final reconstruction quality. The multiplex cameras and the posterior scaling also improve the reconstruction quality, stabilizing the training and preventing cameras to get stuck in local minima. Visual comparisons of the specific ablations are available in the supplements.

**Applications.** One of the contributions of this work is the extraction of explicit mesh with material properties from the learned neural reflectance volume. The process is described in the supplementary material. The resulting mesh can be realistically placed in an Augmented Reality (AR) scene or in a 3D game. In addition, one could edit the BRDF materials on the recovered mesh. See Fig. 1 for sample results of these applications, where our recovered 3D assets blend well in a given 3D scene.

**Limitations.** SAMURAI achieves large strides in the decomposition of in-the-wild image collections compared to prior art. However, we still rely on rough pose initialization. GNeRF proposes a reconstruction technique without any pose initialization but it fails on the challenging in-the-wild datasets. Furthermore, SAMURAI produces slightly blurry textures. This is especially noticeable in the cape scene in Fig. 8. Here, the cape has a repeating, high-frequency texture. Reconstruction of this high-frequency texture requires near-perfect camera poses. Since this dataset is in a single location and illumination, COLMAP-based pose estimation outperforms SAMURAI based pose alignment. However, SAMURAI enables the reconstruction of highly challenging datasets of online image collections where COLMAP completely fails. Our BRDF and illumination decomposition is also not capable of modelling shadowing and inter-reflections. As we mainly tackle object decomposition, the shadows and inter-reflections are not crucial. Removing the need for pose initialization along with modeling shadows and inter-reflections form an important future work.

## 5 Conclusion

SAMURAI is a carefully designed optimization framework for joint camera, shape, BRDF, and illumination estimation. It can work on in-the-wild image collections captured in varying backgrounds and illuminations and with different cameras. Results on existing and our new challenging dataset demonstrate good view synthesis and relighting results, where several existing techniques fail. In addition, our mesh extraction allows the resulting 3D assets to be readily used in several graphics applications such as AR/VR, gaming, material editing *etc*.

## Acknowledgments and Disclosure of Funding

This work has been partially funded by the Deutsche Forschungsgemeinschaft (DFG, German Research Foundation) under Germany's Excellence Strategy – EXC number 2064/1 – Project number 390727645 and SFB 1233, TP 02 - Project number 276693517. It was supported by the German Federal Ministry of Education and Research (BMBF): Tübingen AI Center, FKZ: 01IS18039A.

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
