# *Supplementary Material for*
# SAMURAI: Shape And Material from Unconstrained Real-world Arbitrary Image collections

**Mark Boss**
University of Tübingen
Google

**Andreas Engelhardt**
University of Tübingen
Google

**Abhishek Kar**
Google

**Yuanzhen Li**
Google

**Deqing Sun**
Google

**Jonathan T. Barron**
Google

**Hendrik P. A. Lensch**
University of Tübingen

**Varun Jampani**
Google

In the supplement we first introduce additional details in Section A1, A2, and A3. The mesh extraction is described in Section A4. We summarize different datasets used in the experiments in Section A5 and state the categorization per dataset. Lastly, we show more experiments and results of SAMURAI in Section A6. For an overview of this work and further results, please also consider visiting our project page: `https://markboss.me/publication/2022-samurai/`.

## A1  Random Offset Annealed Fourier Encoding

The Fourier Encoding $\gamma : \mathbb{R}^3 \mapsto \mathbb{R}^{3+6L}$ used in NeRF [7] encodes a 3D coordinate $\boldsymbol{x}$ into $L$ frequency basis:

$$\gamma(\boldsymbol{x}) = (\boldsymbol{x}, \Gamma_1, \ldots, \Gamma_{L-1}) \tag{1}$$

Where each frequency is encoded as:

$$\Gamma_k(\boldsymbol{x}) = \left[\sin(2^k \boldsymbol{x}), \cos(2^k \boldsymbol{x})\right] \tag{2}$$

BARF [5] and Nerfies [8] introduced annealing of the Fourier Frequencies using a weighting:

$$\Gamma_k(\boldsymbol{x}; \alpha) = w_k(\alpha) \left[\sin(2^k \boldsymbol{x}), \cos(2^k \boldsymbol{x})\right] \tag{3}$$

$$w_k(\alpha) = \frac{1 - \cos\left(\pi \operatorname{clamp}(\alpha - k, 0, 1)\right)}{2} \tag{4}$$

where $\alpha \in [0, L]$. This can be seen as a truncated Hann window. One downside of this form of encoding is that all frequencies are axis-aligned. In Tancik *et al.* [11] the benefits of adding random frequencies are demonstrated. However, combining this with the sliding cosine window is not easily possible. Therefore, we propose to add random Gaussian offsets $\boldsymbol{R} \in \mathbb{R}^{L \times 3}$ to the frequencies. The offsets $\boldsymbol{R}$ are sampled from $N(0, 0.1)$. This can be thought of as randomly rotating each frequency band:

$$\Gamma_k(\boldsymbol{x}; \alpha) = w_k(\alpha) \left[\sin(2^k \boldsymbol{x} + 2^k \boldsymbol{R}_k), \cos(2^k \boldsymbol{x} + 2^k \boldsymbol{R}_k)\right] \tag{5}$$

36th Conference on Neural Information Processing Systems (NeurIPS 2022).

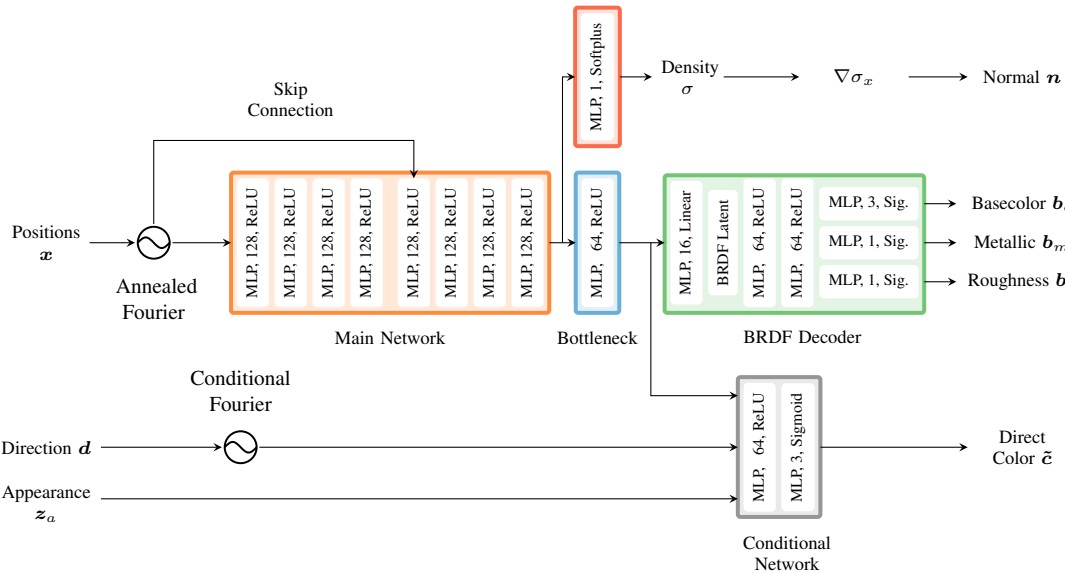

Figure A1: **Architecture.** The detailed architecture of our network. Note that the conditional network and the direct color is only used in the early stages of the training for stabilization. It does not contribute to the final decomposition result. Our main outputs include the Density $\sigma$, the normal $n$ and the BRDF ($b_c$, $b_m$, $b_r$), which are used for rendering our actual output color $\hat{c}$ with Neural-PIL [2].

## A2 Weight Updating for Camera Multiplex

As we stochastically sample points for each batch, a potential bad camera can have favorable samples and outperform a better camera. We alleviate this issue by storing the weights for each of our optimization images in a memory bank $W \in \mathbb{R}^{j \times 4}$. These can then be updated during the optimization and reduce the impact of the sample distributions. Furthermore, we store a memory bank of velocities $V \in \mathbb{R}^{j \times 4}$ to speed up the selection of the best camera pose. The weight matrix is then updated with the new weights $S_j$ using:

$$W_j^* = \max(W_j + mV_j^* + g, 0) \tag{6}$$
$$V_j^* = m * V_j + g \tag{7}$$
$$g = s(S_j - W_j) \tag{8}$$

where the new weights $W_j^*$ and velocities $V_j^*$ replace the old ones, the parameters $m$ represent the momentum, and $s$ the learning rate. The values for these are $0.75$ and $0.3$, respectively.

## A3 Network Architecture and Further Training Details

The input images for our network are used without cropping. We sample the foreground area thrice as often as the background regions to circumvent the potential large background areas. As the resolution varies drastically and can be large, we further resize the images so that the largest dimension is 400 pixels.

The detailed configuration of our network is shown in Fig. A1. We use 10 Random Offset Annealed Fourier Frequencies for the positional encoding. These are annealed over 50000 steps using Eq. 4. The directions are encoded using 4 non-annealed and non-offset Fourier frequencies. The losses in section 3.2 - **Overall network and camera losses** of the main paper are weighted with the following scalars besides the optimization schedule scalars $\lambda_b$, $\lambda_a$:

| $\lambda_{\text{ndir}}$ | $\lambda_{\text{Smooth}}$ | $\lambda_{\text{Dec Sparsity}}$ | $\lambda_{\text{Dec Smooth}}$ |
|---|---|---|---|
| 0.005 | 0.01 | 0.01 | 0.1 |

The coarse-to-fine optimization is further governed by $\lambda_c$. This parameter mainly interpolates between the available resolution of the largest dimension from 100 to 400 pixels and the number of cameras from 4 to 1. The softmax scalar $\lambda_s$ is also driven by $\lambda_c$ and fades from a scalar of 1 to 10.

Furthermore, we apply gradient scaling to the gradients for the network by the norm of 0.1. The camera gradients are not clipped or scaled.

## A4 Mesh extraction

Similar to NeRD [1], we perform a mesh extraction from the learned reflectance neural volume. However, we differ from their method. In the first step, we perform a marching cubes extraction step similar to the one proposed in NeRF [7]. However, as the naive marching cube algorithm can have block artifacts, we sample 2 million points on the mesh surface and cast rays toward the surface. The resulting point cloud is converted to a refined mesh using Poisson reconstruction. This refined mesh provides more details and smoother surfaces. We then UV unwrap the resulting mesh in Blender's [3] automatic UV unwrapping tool and bake the world space positions into the texture map. We can then query all surface locations in our fine network and compute the BRDF texture maps. We then save the textured mesh in the GLB format for easy deployment. The extraction of a mesh takes around 3-5 minutes.

## A5 Additional Details of Datasets

| Scene | Multi-Illumination | Known Poses | Notes |
|---|---|---|---|
| Gold Cape | ✗ | ✓ | From NeRD [1] |
| Head | ✗ | ✓ | From NeRD [1] |
| Syn. CarWreck | ✓ | ✓ | Synthetic from NeRD [1] |
| Syn. Globe | ✓ | ✓ | Synthetic from NeRD [1] |
| Syn. Chair | ✓ | ✓ | Synthetic from NeRD [1] |
| Mother Child | ✓ | ✓ | From NeRD [1] |
| Gnome | ✓ | ✓ | From NeRD [1] |
| Statue of Liberty | ✓ | ✗ | Online collection |
| Chair | ✓ | ✗ | Online collection |
| Duck | ✓ | ✗ | Self-collected |
| Fire Engine | ✓ | ✗ | Self-collected |
| Garbage Truck | ✓ | ✗ | Self-collected |
| Keywest | ✓ | ✗ | Self-collected |
| Pumpkin | ✓ | ✗ | Self-collected |
| RC Car | ✓ | ✗ | Self-collected |
| Robot | ✓ | ✗ | Self-collected |
| Shoe | ✓ | ✗ | Self-collected |

Table A1: **List of Datasets.** List of all datasets and the classification into multi-illumination and known poses.

A list of our datasets used in the evaluation is shown in Table A1. Our new dataset in the last section of Table A1 consists of around 70 images each. We tried to replicate the online collection setting as much as possible by using different cameras (Pixel 4a, iPhone 7 Plus, Sony alpha 6000), capturing the objects in different unique environments, and replicating the hand-held capture setup with varying distances. Even with the extensive manual tuning of parameters, we cannot estimate the camera poses in traditional methods such as COLMAP [9, 10]. Fig. A2 shows an overview of the images in two image collections.

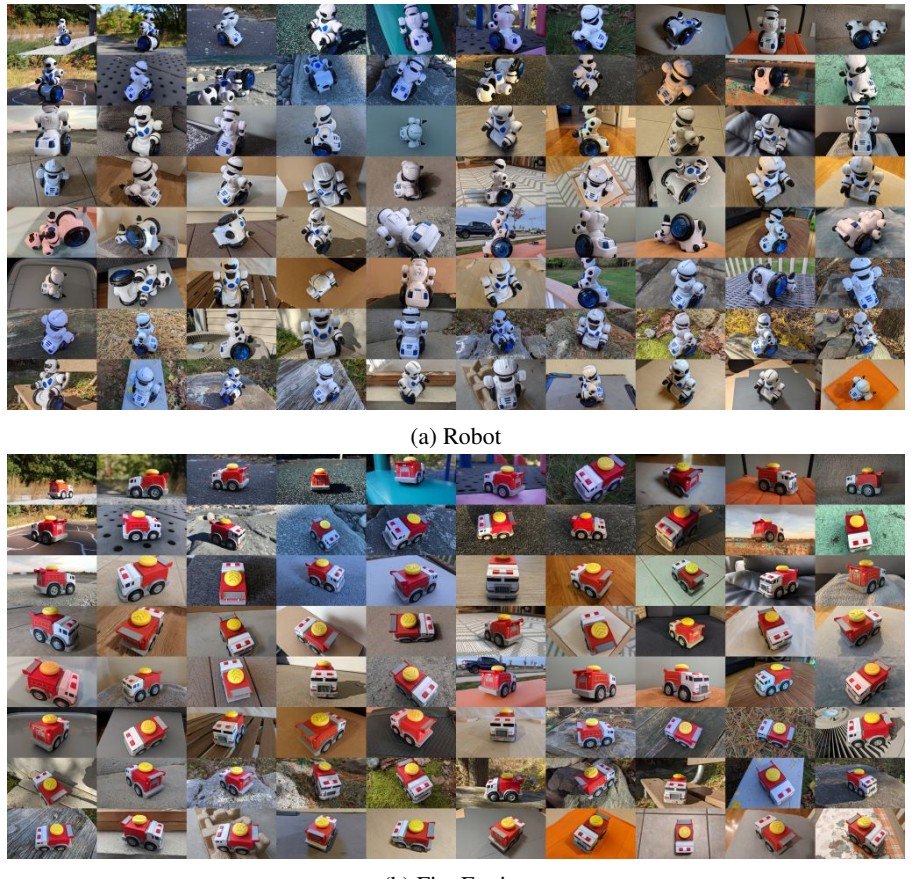

(a) Robot

(b) Fire Engine

Figure A2: **Dataset Overview.** Notice the complex illumination conditions and the drastically varying locations. Also, the distances vary quite severely.

| GT | w/o Regular-ization | w/o Random Fourier Offsets | w/o Coarse 2 Fine | w/o Posterior Scaling | w/o Camera Multiplex | Full |
|---|---|---|---|---|---|---|

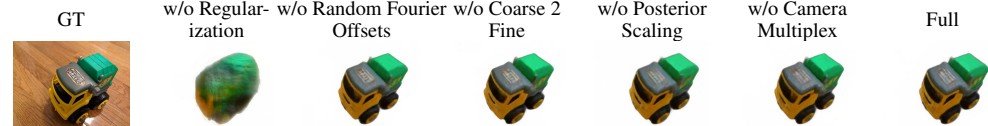

Figure A3: **Visual Ablation.** Each of our novel additions improve the reconstruction. In this particular scene the regularization is critical for the decomposition. The coarse 2 fine, posterior scaling and camera multiplex ablations mainly have a reduced sharpness in the sticker on the top. In the random Fourier offset annealing striping patterns are apparent in the some areas which are alleviated with our full model.

## A6 Additional Experiments

**Ablation.** In Fig. A3 the result of our ablation study is shown. The benefit of our regularization is easily apparent in this scene. Furthermore, our coarse-to-fine, posterior scaling, and camera multiplex help recover slightly sharper details but especially help stabilize the optimization. The random Fourier offsets also alleviate some slight striping artifacts.

**Visual Results.** In Fig. A4, we show additional results of SAMURAI on several highly challenging, multiple illumination scenes. Our method can create plausible decomposition, which produces convincing results when re-rendered in unseen views and illumination conditions. Even fine details, like the RC Car's antenna, are preserved well. Only the legs of the chair object are not reproduced well. However, the legs are also not detected well by our automatic segmentation with U2-Net.

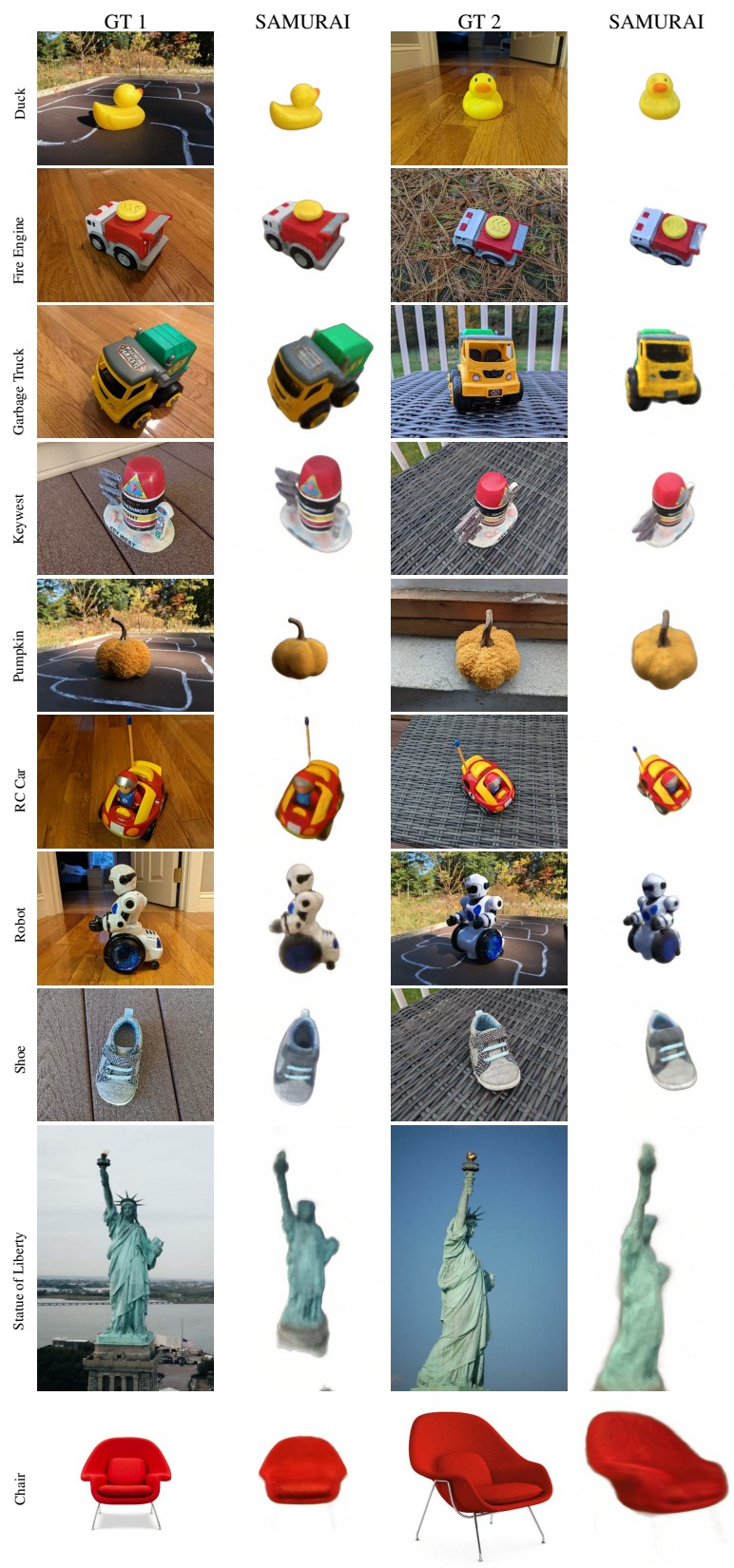

Figure A4: **Additional Results.** Renderings with camera poses and illumination from test views demonstrate plausible novel view synthesis and re-lighting on various datasets.

**NeRS modifications.** The default implementation of NeRS [12] does not implement mini-batching. This means all images are optimized simultaneously in a resolution of $256 \times 256$. This works well for a few images, but the GPU memory runs out with larger image collections. We have created a modified version that implements mini-batching for a fair comparison. Still, NeRS is only capable of working on single illumination datasets.

**Procrustes Analysis of Camera Poses.** We evaluate the quality of the reconstructed camera poses against a reference obtained from COLMAP [9, 10]. References are only available for the scenes of the NeRD dataset. First, we align the camera locations using Procrustes analysis [4] as in [5]. The rotation error is reported as a mean deviation in degrees, while the translation error is computed as the mean difference in scene units of the reference scene. In contrast to the evaluation of the view synthesis and rendering performance, we here use all cameras from the training data for comparison like it has been done in concurrent works [5, 6].

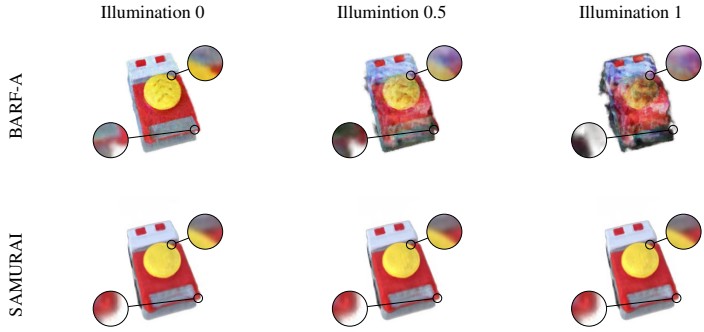

Figure A5: **BARF-A appearance conditioning entanglement.** BARF-A conditions the radiance on the view direction and the trainable appearance embedding. We interpolate between two illuminations for our method and BARF-A. We can see that the illumination is entangled with the pose. Here, we provide magnifications on edges where this entanglement is noticeable. Due to different camera coordinate systems and optimizations, we provide similar views from SAMURAI. Here, it is clear that in SAMURAI, the texture remains static.

**Illumination Conditioning Entanglement in BARF-A** In the main paper, we showed the entanglement of pose and view direction conditioning with BARF. In BARF-A, the illumination is additionally modeled as an appearance embedding. This results in an even higher ambiguity in representing high-frequent surface details. The ambiguity is visible when we interpolate between two appearance embeddings in Fig. A5. The texture is also shifting based on the appearance embedding and even distorts severely for the specific view indirection. This is primarily due to the large baseline between the camera of 'Illumination 1' and 'Illumination 0'. In SAMURAI, our BRDF representation is not only view-independent but also illumination-independent. Entanglements, as seen in BARF or BARF-A, are impossible with our method. This enables our higher-quality camera and asset reconstructions.

| Translation Error | Rotation deg Error | PSNR↑ |
|---|---|---|
| 0 | 0 | 29.48 |
| 0.1 | 0.1 | 20.21 |
| 0.5 | 0.5 | 16.89 |
| 1 | 1 | 12.58 |
| 3 | 3 | 9.34 |
| 5 | 5 | 0.98 |

Table A2: **Performance of Neural-PIL with varying inaccurate poses.** The performance of Neural-PIL quickly degrades on even mild pose inaccuracies. SAMURAI is robust to performance inaccuracies due to the posterior scaling and achieves a PSNR of 23.94 dB when starting from the quadrant-based poses.

**Comparison with Neural-PIL with noisy camera poses** Under the assumption that poses are recoverable, but the poses will be slightly noisy, we can show the performance degradation of Neural-PIL [2] in Table A2. If our method leverages GT poses without optimization, Samurai (ours) obtains similar results as Neural-PIL. Our method is mostly a generalization of Neural-PIL to also jointly optimize camera poses. In Table A2, it is clear that Neural-PIL degrades severely even under slightly noisy poses. Samurai achieves a PSNR of 23.84 dB. So even with minor noisy poses, our method outperforms Neural-PIL significantly. It is worth noting that SAMURAI starts from the rough quadrant-based poses, which are not close to the GT pose. Even if our method does not achieve the same reconstruction performance as Neural-PIL with known poses - the difference is not too far off.