# OpenReview forum: "SAMURAI: Shape And Material from Unconstrained Real-world Arbitrary Image collections"
_NeurIPS.cc/2022/Conference — NeurIPS 2022 Accept_

### Official Review · Reviewer_m5j4 · 2022-07-11

**Rating:** 6
**Confidence:** 4
**Soundness:** 3 good
**Presentation:** 3 good
**Contribution:** 2 fair

**Summary:**

This paper proposes a NeRF-based single-object inverse rendering pipeline where the camera intrinsics is unknown and extrinsics are coarsely initialized. The neural implicit based reflectance (BRDF) is also optimized separately for relighting/material editing. Compared with previous methods that uses camera intrinsics and extrinsics estimated from a third-party pipeline (eg. COLMAP), the proposed method is more robust in cases where COLMAP fails to estimate accurate poses due to the camera optimization module. A small dataset consisting of images captured with different background/illumination for 8 objects is collected and used in evaluation of the proposed method. The experiment results show that the proposed method outperforms the compared state-of-the-art methods that also use unposed input images.


**Questions:**

Please see the questions in the weakness section.

**Limitations:**

The method does not specifically handles shadows, as the author pointed out in the paper.


**Strengths And Weaknesses:**

Strength

+The joint optimization of the BRDF, occupancy represented as neural implicit functions and the camera parameters (poses and intrinsics) given unposed images captured under different illuminations/backgrounds is a challenging task. The recipes used in this submission can be of good reference for the following works that pushes the NeRF-based inverse rendering towards in-the-wild scenarios.

+Compared with the similar method (BARF), the performance boost of the proposed pipeline is obvious.

+A small dataset is proposed for challenging cases where the COLMAP fails (L276-L277). This dataset can be useful for testing algorithms using unposed images with varying illumination as inputs.


Weakness

-My main concern is about the limitation in the cases where the proposed method performs better than the compared method:
Other methods, such as NeRF-W, also uses internet photos with varying background/illumination/camera intrinsics as inputs; the key difference between the work and the previous works is that this work is robust to cases where COLMAP fails to estimate good camera poses or does not work at all, versus the assumption that COLMAP works for in-the-wild dataset in other methods. Although this work has proposed a small dataset on which COLMAP is not working at all and shows that the proposed method works on this dataset, the following question is not answered: if COLMAP does estimate some noisy camera poses, can this method with optimized pose method still output perform the counterparts (either the other parts of this method + COLMAP pose, or NeRD + COLMAP pose) ? From Tab.3 of this paper, it shows that if COLMAP generates good camera pose, then the performance of the proposed method is not as good as the compared methods. This seems to reduce the application scope of this proposed method: it only works better if COLMAP does not work, which is not always the case given that COLMAP has decent robustness.

-Self occlusion/shadowing is not considered. This poses a challenge for objects with concave shapes (grooves) that generate cast shadows, which may be reconstructed as albedo variations.

-Based on L181-L199, the 'valid' sampling ranges for the rays are determined by the intersections between the predefined object-centric sphere and the camera rays, which are intern determined by the camera pose and intrinsics. As a result, the sampling location along the rays depends on the camera intrin/extrin. If that is the case, how the dependence is dealt with during the optimization process is not well explained.

-The weighting scheme in camera multiplexes depends on the reliability of each candidate pose. How about the cases where none of the positions from one multiplex is reliable? Are the corresponding images ignored during reconstruction?

---

> ### Author Response · Authors · 2022-08-02
> **Response to m5j4**
>
> **Comparison with noisy camera poses**: Please see the answer in the main response above.
>
> **Reconstruction quality wrt. existing works**: Please see the answer in the main response above.
>
> **Performance when poses available**: Please see the answer in the main response above.
>
> **Sampling range during optimization**: Our sampling bounds stay fixed during the optimization process at the origin of the coordinate system. We enforce that our cameras look roughly toward that origin. The cameras can move freely, and we simply calculate the intersection points of an imaginary ray towards the origin (Shown in Fig. 3 in a dashed line). The distances toward the intersections can then be transferred to the actual rays, which generates a view frustum. Therefore, we only sample in a predefined area during the optimization.
>
> **Reliability of camera pose initialization**: It can indeed occur that all candidate poses are equally bad. However, it is unlikely that all images will have such poor initialization. Therefore, we introduced an image level posterior scaling (L215-L221), where we compare the weighted losses against all other views. If all poses are bad, we reduce the backpropagation towards the network for the views. Similar to the Camera Multiplex, the camera poses can still improve unhindered. Therefore, specific images are ignored if the poses do not improve during training.

---

### Official Review · Reviewer_hfWC · 2022-07-11

**Rating:** 6
**Confidence:** 4
**Soundness:** 3 good
**Presentation:** 4 excellent
**Contribution:** 3 good

**Summary:**

This paper proposed a method that works on in-the-wild image collections of an object to estimate the shape, BRDF, per-image camera pose, camera intrinsics and illumination in a jointly optimized framework.
This problem is very challenging and under constraint as the camara parameters, illuminations and even the background of the images may vary a lot across the online image collections. To my knowledge, this is the first method that aims at solving the shape, BRDF and camera poses with this challenging setup.

The paper proposed several components that based on the recent Neural-PIL-Rendering technique to construct the entire pipeline, including: an object-centric camera parametrizetion to learn the clipping planes per image; camera multiplex optimization to avoid local minima during the optimization; posterior scaling of input images to suppress the influence of corrupts images; a two-stage mesh extraction for refined mesh reconstruction.

**Questions:**

See weaknesses part.

**Limitations:**

The most of the limitations are discussed in the paper.

**Strengths And Weaknesses:**

Strengths
- The paper is well written and easy to follow. The figures, tables and videos help the understanding of the paper and identify its contributions.

- The proposed camera multiplexes, although been used in mesh optimization works [22] before, is the first time being used in the context of neural volume rendering. This technique will dynamic re-weighing the each camera loss to reduce the influence of bad camera poses during the optimization of the shape and materials.

- The proposed posterior scaling of input images also makes sense in the context of suppressing the influence of corrupted images. This technique can also be used in a broader application: when dealing with online image collections and images with poor quality can be dynamically re-weigh.

- The evaluation of the method over real and synthetic datasets demonstrates the effectiveness of this method in recovering shape and material under unknown camera poses, changing illuminations and varying backgrounds. (From Table 1 and Table 3, Table A3)

- The Ablation study in table 2 also validates the effectiveness of each component of the methods.

Weaknesses
- In table 1, the SAMURAI's performance is lower when the camera poses are available. Why?

- I have a concern about the quality of the reconstructed meshes. In Figure A4, the reconstructed mesh of "Statue of Liberty" and "Chair" is inaccurate, and the texture is also blurry.

- The novel view synthesis quality is very low compared to the prior "fix-illumination" or "known camera pose" methods. As can be seen in Table 1, Table 3 where the reconstructed images have much lower PSNR compared to prior works.

---

> ### Author Response · Authors · 2022-08-02
> **Response to hfWC**
>
> **Performance when poses available**: We do not use the poses in our method and always use the rough pose initialization. We will change the wording to clarify that our method still leverages the direction-based poses, not the GT poses. We split the datasets to enable comparison with NeRD and Neural-PIL for the datasets where poses are available.
>
> **Reconstruction quality on internet collections**: The Internet image collections provide additional challenges, as the Statue-of-Liberty scene consists of a collection of images under highly varying capture scenarios. The shots consist of drone images, images from ships, directly under the statue, from helicopters, the mainland, etc. Here, the focal lengths and distances can vary extremely starkly. For the chair scenes, the automated U2-net masking did not always include the legs, which are distinct features for pose alignment. Our SAMURAI dataset mostly captures some of these challenges (varying camera and distances) but more constrainedly.
>
> **Reconstruction quality wrt. existing works**: Please see the answer in the main response above.

---

### Official Review · Reviewer_eZCF · 2022-07-17

**Rating:** 6
**Confidence:** 3
**Soundness:** 3 good
**Presentation:** 4 excellent
**Contribution:** 3 good

**Summary:**

This paper uses a neural field-based approach to estimate the shape, BDRF parameters, and per-image camera pose and illumination from the in-the-wild image collections. It is the first method that is able to estimate all parameters simultaneously. Conventional neural field-based approaches require an almost-correct camera pose to estimate the shape and BDRF parameters. The proposed method replaces the need for accurate camera pose with a rough user-initialized camera pose quadrant. This paper introduces a novel in-the-wild image collection dataset in which COLMAP is hard to estimate the camera pose. Experimental results show that the proposed method achieves much better accuracy in the challenging dataset (pose not known).

**Questions:**

- In L157, the authors claim that they do not use the coarse-to-fine network. What if the proposed method use coarse network? Wouldn’t the accuracy be better? In addition, what is the coarse-to-fine optimization in the ablation study?
- Why couldn’t the proposed method achieve GT camera pose? What if the proposed method utilizes deep camera pose estimation (such as Wide-Baseline Relative Camera Pose Estimation with Directional Learning) to improve the accuracy? Would it be possible?


**Limitations:**

Yes, the authors has adequately addressed the limitations.

**Strengths And Weaknesses:**

# Strengths

- The proposed method is able to solve a highly under constrained problem with rough camera pose initialization. It estimates the BRDF parameters, unit-length surface normal, volume density, latent-per-image illumination vectors, per image camera pose and intrinsics.
- Flexible camera parameterization for varying distances. As the proposed method targets for in-the-wild image collection, the near and far bound of conventional methods cannot be applied. Thus, the proposed method places the camera in a distance where the is visible given the field of view.
- Camera multiplex optimization. It is challenging to optimize the camera pose due to local minima. The proposed methods optimize multiple camera poses with their corresponding weights based on the camera loss to find the best possible camera pose.
- Posterior scaling of input images. Similar to camera multiple optimization, the proposed method also optimizes the image collection used for training. It gives different weights based on the noise level for each image.
- The proposed method can be applied for various applications, such as AR & VR, which makes it easy for those applications to insert the real-world objects without huge efforts.


# Weaknesses

- The authors claim that standard pose estimation techniques fail on the challenging images, but there is no justification whether the proposed dataset is a challenging dataset. Proof of pose of estimation failure should be included.
- While the conventional methods automatically estimate the camera pose with COLMAP, the proposed method requires user interaction for each dataset to roughly annotate the position. However, there is no justification why the proposed rough pose estimation is preferred. In addition, there is no ablation study of the camera pose initialization using 3 simple binary questions.
- The intention of the proposed method is to do proof-of-concept of large-scale set objects. However, the qualitative evaluation of large-scale set objects is unavailable in the main manuscript. It is questionable whether the proposed method can be scaled to large-scale set objects. In the supplementary material, there are two online image collections, but the results are not satisfying. In addition, comparison with state-of-the-art methods is unavailable.
    - As in Modeling the World from Internet Photo Collections, the camera pose of large objects images might be estimated using correspondences.

---

> ### Author Response · Authors · 2022-08-02
> **Response to eZCF**
>
> **No proof of COLMAP not working**: In our novel dataset and the internet image collections, we found that COLMAP fails in the correspondence matching step. No reconstruction took place. We have followed best practices and tried extensive parameter tuning for these scenes.
>
> **Need for pose annotation**: We note that objects are often symmetric in a specific plane. For example, a car is mostly symmetric left to right. When the camera is initialized on the back of the car but should be in the front, it is unlikely that a gradient-based optimization will be capable of moving the camera there as the loss would be higher for the side views. We found that happening often. Therefore, we followed recent research such as NeRS [63], which requires a rough camera direction.
>
> **On large-scale objects**: Our method is mostly designed for smaller objects. This is mainly due to the non-global illumination model. Furthermore, occlusions in large-scale objects occur, which we do not model. We leave these challenging points for future work, and the main goal for SAMURAI is tackling small to medium-sized objects.
>
> **No comparison with prior art**: We want to highlight that we compare with recent papers (GNeRF, BARF) and even provided a modified version (BARF-A), which handles varying illumination. These are state-of-the-art methods in camera and neural field reconstruction. Snavely et al. 2007 leverage correspondences, which will fail in our datasets (See **On no proof of COLMAP working**). Our method is not designed for large-scale objects (See the previous answer)
>
> **Removal of coarse network**: We indeed do not use hierarchical sampling as in NeRF. We found a general problem with two separate networks: As the camera poses are moving and the networks are fully disjointed, we noticed that instabilities in the camera optimization occur. We, therefore, only use a single network with stratified sampling.
>
> **Coarse-to-fine in ablation**: The coarse-to-fine optimization is the BARF-style Fourier annealing coupled with the resolution increase during the optimization. We will clarify that in the revision.
>
> **Achieving GT pose estimation**: Few methods arrive at the correct GT pose during optimization, and the current state-of-the-art methods in the neural field and camera pose estimation also do not achieve this. In general, the joint reconstruction of intrinsic, extrinsic, and shape estimation is highly challenging, especially when no correspondences are available (See section on COLMAP).
>
> **Leveraging initialization**: Most pose estimation methods require either a video, fixed backgrounds, correspondences, or are category specific. This is not the case for online image collections of objects.

---

### Author Response · Authors · 2022-08-02
**General Response**

First, we want to thank the reviewers for recognizing that our method is capable of solving “a highly under constrained problem” (eZCF, hfWC) with several technical novel contributions (eZCF, hfWC, m5j4). Each contribution shows an effect in the ablations (hfWC), and the overall method provides an obvious boost compared to the prior art (hfWC). In this general response, we address common questions by reviewers. .

**Reconstruction quality w.r.t. existing works** (hfWC, m5j4): We agree that the quality is lower than prior works leveraging near GT poses (NeRD or Neural-PIL). But, we tackle a significantly more challenging problem of jointly estimating camera poses, shape reconstruction, and material decomposition. Especially in datasets where objects are located in different locations and illuminations, traditional pose reconstruction methods fail. We show the influence of noisy poses in the **Comparison with noisy camera poses** below.

**Comparison with noisy camera poses** (hfWC, m5j4): Under the assumption that poses are recoverable, but the poses will be slightly noisy, we can show the performance degradation of Neural-PIL in Tab. T1. If our method leverages GT poses without optimization, Samurai (ours) obtains similar results as Neural-PIL as our method is mostly a generalization of Neural-PIL to also jointly optimize camera poses. In Tab. T1, it is clear that Neural-PIL degrades severely even under slightly noisy poses. Samurai achieves a PSNR of 23.84 dB. So even with minor noisy poses, our method outperforms Neural-PIL significantly. It is worth noting that SAMURAI starts from the rough quadrant-based poses which are not close to the GT pose. Even if our method does not achieve the same reconstruction performance as Neural-PIL with known poses - the difference is not too far off.

| Translation % Error | Rotation ° Error | PSNR  |
|---------------------|------------------|-------|
| 0                   | 0                | 29.48 |
| 0.1                 | 0.1              | 20.21 |
| 0.5                 | 0.5              | 16.89 |
| 1                   | 1                | 12.58 |
| 3                   | 3                | 9.34  |
| 5                   | 5                | 0.98  |

**Tab T1.** - Performance of Neural-PIL with varying inaccurate poses. SAMURAI achieves a PSNR of 23.94 dB when starting from the quadrant-based poses.

---

> ### Author Response · Authors · 2022-08-09
> **Discussion**
>
> Dear Reviewers, Thanks for your constructive feedback. We hope to have clarified most of the reviewer questions in our response. As we are nearing the end of the author-reviewer discussion period, we would like to give a gentle reminder in case you have any more questions or concerns.

---

### Meta-Review · Area_Chair_jDWS · 2022-08-31

**Recommendation:** Accept
**Confidence:** Certain

**Metareview:**

This paper had notable consistent reviews.  All reviews were thoughtful, and there was a consensus that this paper tackles an important problem in a way that has not been explored.  While there were some weaknesses highlighted in the review process, discussion and the author rebuttal ameliorated all major concerns.  Therefore I am accepting this paper.

**Award:**

No

---

### Decision · Program_Chairs · 2022-09-14

Accept